# Upregulation of miR145 and miR126 in EVs from Renal Cells Undergoing EMT and Urine of Diabetic Nephropathy Patients

**DOI:** 10.3390/ijms232012098

**Published:** 2022-10-11

**Authors:** Veronica Dimuccio, Linda Bellucci, Marianna Genta, Cristina Grange, Maria Felice Brizzi, Maddalena Gili, Sara Gallo, Maria Laura Centomo, Federica Collino, Benedetta Bussolati

**Affiliations:** 1Department of Molecular Biotechnology and Health Sciences, University of Turin, 10124 Turin, Italy; 2Laboratory of Translational Research in Paediatric Nephro-Urology, Fondazione Ca’ Granda IRCCS Ospedale Maggiore Policlinico, 20122 Milan, Italy; 3Department of Medical Sciences, University of Turin, 10124 Turin, Italy; 4Department of Clinical Sciences and Community Health, University of Milan, 20122 Milan, Italy

**Keywords:** diabetic nephropathy, urinary extracellular vesicles, miRNAs, epithelial to mesenchymal transition, biomarkers

## Abstract

**Simple Summary:**

Diabetic nephropathy is one of the most frequent complications of diabetes, resulting from diffuse damage to different kidney cells. The identification of subjects at risk is mandatory to prevent its development and provide appropriate therapies reducing the unmanageable evolution towards end-stage kidney disease. The aim of this work was to identify urinary-derived extracellular vesicles (EVs) miRNA cargo to be used as biomarker of kidney damage in diabetic patients. The miRNA profile was then correlated with the molecular mechanism associated with the glomerular and tubular damage using a diabetic-like model. In patients, miR145 and miR126 in urinary EVs increased together with albuminuria. MiR145 and miR126 increased in parallel in EVs from renal epithelial cells undergoing transition to a fibrotic mesenchymal phenotype. These data unveiled a role for miR126 and miR145 as the biomarkers of damage progression and proteinuria development in diabetic nephropathy.

**Abstract:**

Diabetic nephropathy (DN) is a severe kidney-related complication of type 1 and type 2 diabetes and the most frequent cause of end-stage kidney disease. Extracellular vesicles (EVs) present in the urine mainly derive from the cells of the nephron, thus representing an interesting tool mirroring the kidney’s physiological state. In search of the biomarkers of disease progression, we here assessed a panel of urinary EV miRNAs previously related to DN in type 2 diabetic patients stratified based on proteinuria levels. We found that during DN progression, miR145 and miR126 specifically increased in urinary EVs from diabetic patients together with albuminuria. In vitro, miRNA modulation was assessed in a model of TGF-β1-induced glomerular damage within a three-dimensional perfusion system, as well as in a model of tubular damage induced by albumin and glucose overload. Both renal tubular cells and podocytes undergoing epithelial to mesenchymal transition released EVs containing increased miR145 and miR126 levels. At the same time, miR126 levels were reduced in EVs released by glomerular endothelial cells. This work highlights a modulation of miR126 and miR145 during the progression of kidney damage in diabetes as biomarkers of epithelial to mesenchymal transition.

## 1. Introduction

Diabetic nephropathy (DN) is one of the most frequent and severe chronic complications of diabetes, characterized by persistent high albuminuria and a subsequent decline in the glomerular filtration rate [1,2]. The changes in kidney function are associated with specific histopathological findings in glomerular and tubulointerstitial compartments, with renal cell hyperplasia and hypertrophy, the thickening of glomerular and tubular basement membranes and the expansion of tubulointerstitial and mesangial compartments [3,4]. It is well established that patients suffering from both type 1 and type 2 diabetes develop nephropathic complications very early in the progression of the disease; thus, the identification of subjects at risk of DN is required to provide appropriate therapy and slow down evolution towards end-stage renal disease [5]. Currently, the assessment of DN is based on the levels of retention markers, such as creatinine and urea, and on the amount of proteins found in urine. Indeed, albuminuria is today one of the most important indexes applied to assess the progression of DN and one of the main prognostic factors for renal interstitial fibrosis [6]. On the other side, the degree of proteinuria does not accurately reflect the severity or the prognosis of DN, nor is it specific, being a hallmark of glomerular dysfunction [6]. In addition, tissue damage and the induction of inflammation have already occurred by the time that albuminuria is detectable. Finally, a decline in kidney function in diabetic patients can occur in the presence of non-albuminuric or non-proteinuric DN [7], making it difficult to establish the correct timing to initiate the appropriate therapeutic intervention. This intensifies the need for sensitive non-invasive biomarkers of DN [5].

Extracellular vesicles (EVs) are small particles composed of a lipid bilayer secreted by all cell types under both physiological and pathological conditions. EVs display a relevant role in cell signaling and communication, thanks to the potential to transfer their molecular cargo (proteins, lipids and nucleic acids) and play a central function in kidney physiology and pathology [8]. In particular, EVs present in urine (uEVs) carry molecules that are characteristic of the epithelial cells present in the whole length of the urinary tract [9]. Accordingly, uEVs are attracting increasing interest as potential urinary biomarkers and may represent a valuable source of information related to the state of renal tissue. Indeed, several data have highlighted the possible use of uEV cargo, including miRNAs, as biomarkers of kidney diseases [8,9], possibly reflecting their dysregulation in the renal tissue at various stages of DN [10]. In fact, miRNAs are relatively stable in tissue and biological fluids, particularly when carried by EVs [11]. However, the diagnostic application of those biomarkers still needs multicenter validation and large patient cohorts [9].

In search of markers of DN progression in preliminary experiments, we here assessed a panel of uEV miRNAs previously related to endothelial injury, tubular damage and renal fibrosis [12,13,14,15,16,17,18,19,20], comparing miRNA levels in diabetic patients stratified based on proteinuria levels. We therefore focused our attention on two miRNAs, miR126 and miR145, that showed an altered expression in patients with DN during disease progression. Moreover, we focused on the identification of the renal cells potentially responsible for the uEV-associated miR release and the biological process behind this phenomenon. In in vitro diabetes-mimicking models generated by hyperglycemic and fibrotic stimulation, miRNA levels released by tubular and glomerular cell-derived EVs correlated with the induction of epithelial to mesenchymal transition (EMT).

## 2. Results

### 2.1. uEV Characterization

EVs were isolated from the urine of 46 diabetic patients following the protocol in [21]. uEVs expressed the classical exosomal markers, tetraspanins (CD63, CD9 and CD81), as evaluated by super resolution microscopy. DN uEVs appeared heterogeneous, expressing single or multiple tetraspanins at single uEV level (Figure 1A). Moreover, the renal origin of uEVs was confirmed by the presence of AQP1 and AQP2, markers of different nephron segments (Figure 1A). TEM analysis reveals the typical size distribution of EVs (Figure 1B) and the typical cap-shaped morphology (Figure 1B inset). Cytofluorimetric analysis confirmed the presence of tetraspanins, as well of classical urinary markers, such as CD24 and CD133 [21] and the epithelial marker CD326. No expression of the endothelial markers CD31 and CD146 was detected (Figure 1C).

uEV were then divided based on the urinary albumin level into the following groups: diabetes patients with normoalbuminuria (NAlb DN), patients with microalbuminuria (MiAlb DN) and patients with macroalbuminuria (MaAlb DN). Table 1 summarizes the clinical features of diabetic patients enrolled in the study. In patients with diabetes, only serum creatinine values appeared to be normally distributed, and the *t*-test showed significant differences among groups (*p* = 0.04). A group-to-group comparison using the Mann–Whitney test for the other non-normally distributed data evidenced a significant difference in glomerular filtration rate and albuminuria but not in glycated hemoglobin values. No significant difference was observed in uEV size between groups (Table 1).

### 2.2. miRNA Modulation in uEVs from Diabetic Patients

In search of markers related to DN progression, uEVs from diabetic patients with normo, micro, or macroalbuminuria were analyzed for the expression of a panels of miRNAs previously reported to be involved in renal cell damage or fibrosis (Appendix A) [12,13,14,15,16,17,18,19,20]. In preliminary experiments, no significant modulation of miR21, miR24, miR221, miR296, and miR320c was observed (Appendix A). On the contrary, the level of miR145 and miR126 in uEVs from diabetic patients appeared significantly modulated within the groups. In particular, a significant increase of miR126 was observed in proteinuric diabetic patients compared to diabetic patients without kidney disease. No differences were detected between micro- and macroalbuminuric patients (Figure 2A). It is of interest that miR126 was previously reported to be elevated in urine of diabetic patients [20]. In parallel, miR145 levels were significantly higher in uEVs from DN patients in respect to normoalbuminuric patients, and further increased with the development of proteinuria (Figure 2B). This result corroborates and extends the observation that uEV miR145 level positively correlates with the onset of microalbuminuria during diabetes [19].

### 2.3. miRNA Modulation in EVs Released by Glomerular Cells in an In Vitro Diabetic Model of Nephropathy

To assess the possible origin of the increased uEV miRNAs, podocytes and glomerular endothelial cells were subjected to TGF-β1 treatment in a millifluidic glomerular system, to mimic complication associated with long-term diabetes, and EVs were isolated separately from the two compartment supernatants (Figure 3A). Under TGF-β1 stimulation, podocytes underwent EMT with a significant increase in α-SMA, N-cadherin, SLUG and TWIST (Figure 3B) with respect to unstimulated podocytes. In the millifluid system, podocytes after TGF-β1 stimulation showed an increased expression of miR145, while miR126 showed no significant changes (Appendix A). Interestingly, a significant increase in both miR145 and miR126 was detected in EVs derived from podocytes subjected to TGF-β1 (Figure 3C). On the contrary, in EVs derived from glomerular endothelial cells, TGF-β1 stimulation was accompanied by a relevant reduction in EV-derived miR126 levels (Figure 3D), whereas miR145 was absent in all the conditions tested (not shown). No significant changes in miR126 or miR145 levels were observed in endothelial cells when treated with TGF-β1 (Appendix A).

### 2.4. miRNA Modulation in EVs Released by Tubular Cells in an In Vitro Model of Diabetic Nephropathy

As proximal tubular cells represent the other potential source of EVs modulated in urine in diabetic conditions, we set up a model of diabetic-induced tubular damage. The tubular cell line HK2 was subjected to glucose and albumin (HSA) overload, to mimic diabetic and proteinuric conditions. As observed in podocytes, renal tubular cells under this condition underwent EMT, showing a significant decrease in the E-cadherin and an increase in α-SMA and TWIST (Figure 4A). No significant modulation of miR126 and miR145 was observed in renal tubular cells subjected to glucose and albumin overload (Figure 4B). At variance, in the EVs released by HK2 submitted to diabetic-induced damage, a significant enhancement of miR145 and miR126 was detected (Figure 4C).

## 3. Discussion

EVs are nanosized particles constantly secreted by all cells and their features reflect the state of the cell of origin, so that they can mirror tissue health and disease [22,23,24]. From a clinical perspective, EVs can be used as appropriate biomarkers to evaluate disease evolution.

In the present study, we show that the uEV levels of miR126 and miR145 change in diabetic conditions during the progression of the renal damage. Moreover, we confirmed the specific increase of those two miRNAs in EVs released by podocytes and not glomerular endothelial cells in diabetic-like conditions under millifluidic perfusion, taking advantage of a 3D glomerular model, as well as by tubular epithelial cells under glucose and protein overload.

Recently, alterations in miRNA expression profiles have been associated with several pathological processes supporting an increasing interest in their exploitation in the diagnosis and prognosis of pathological conditions [11,25]. Here, analyzing several EV associated miRNAs known to be altered in diabetes and DN, we specifically identified two miRNAs, miR126 and miR145, modulated in uEVs during disease progression. MiR126 is known to play an important role in maintaining endothelial cells and angiogenesis, as it is a key regulator of endothelial inflammation and maintains vascular homeostasis [26,27]. In diabetes, miR126 levels were reported to have opposite expression in EVs from urine in respect to those from serum. Indeed, recent findings showed that miR126 levels in uEVs were significantly enhanced in diabetic patients with kidney disease compared to ones without kidney damage [20]. In parallel, miR126 levels were reported as significantly reduced in serum EVs in diabetic conditions [17]. In the present study, we found that miR126 levels were significantly different between uEVs from non albuminuric and uEVs from DN patients with both micro and macroalbuminuria. These results indicate that this marker might be specifically related to the progression of diabetic condition and modulated during the kidney damage. 

The data observed in uEVs from diabetic patients paralleled the results obtained in our in vitro models of diabetes-induced damage. We found that EVs released by podocytes and tubular cells undergoing EMT showed an increase in miR126 levels. In parallel, in our 3D model, glomerular endothelial cells co-cultured in the milli-fluidic glomerular system and submitted to TGF-β1 stimulation showed a marked decrease in miR126 EV level, with no significant changes observed in cells. These data suggest that miR126 packaging in endothelial and renal epithelial cell-derived EVs released in serum can be positively or negatively modulated by pathological stimuli, such as TGF-β1, as well as albumin or glucose overload [17,28]. Our data linking EMT with uEV miR126 levels in renal epithelial cells are supported by previous studies showing that the upregulation of miR126 may promote the development of liver fibrosis in hepatic stellate cells [29]. Moreover, in systemic sclerosis, miR126 has been demonstrated to contribute to the downregulation of the angiogenic factor EGFL7 and may influence fibrosis via collagen modulation [30].

We also found that miR145 levels increased in uEVs from patients with DN with respect to patients with diabetes without renal complication. This result validates in a different cohort the previous observation that in uEVs miR145 positively correlated with the onset of microalbuminuria during diabetes [19]. We here extended this finding in macroalbuminuric subjects in which miR145 was further increased. This result highlights a link for miR145 and disease progression. However, the role of this miRNA in renal tissue damage is unclear, as miR145 can positively or negatively regulate fibrosis in different pathological processes [31,32,33,34,35]. In tumor cells, miR145 prevented cell invasion and EMT [32,33,34]. At variance, in peritoneal fibroblasts, TGF-β1-induced miR145 accounted for EMT induction and fibrosis development through FGF10 decrease [31]. Similarly, in podocytes, miR145 enhancement was recently correlated with foot process effacement and the development of proteinuria [35]. These effects were due to Rho-related pathway targeting and subsequent increase in Rac1 and Cdc42 activity, followed by podocyte injury [35]. In our in vitro experiments, the TGF-β1 treatment of podocytes-induced miR145 expression and determined the release of EVs enriched of this miRNA. In addition, an increase of miR145 was also present in EVs from tubular cells challenged with glucose and albumin overload, in diabetic-like conditions. 

Beside the role as biomarkers, the possible functional effect of EV-contained miR126 and miR145 would be of interest, considering that both miRNAs may affect EMT positively and negatively [26,27,28,29,30,31,32,33,34,35,36]. It could be speculated that uEV cargo during DN may be involved in the amplification of diabetic-induced alterations via EV release or, alternatively, simply represent the modulation of the miRNA packaging in EVs from damaged renal cells. Understanding these intracellular mechanisms and precisely following the axis of miRNA-messenger RNA in kidney cells [23,24,25,36] is crucial for the future use of EVs in the clinical evaluation of DN. Further experiments are needed to clarify this aspect.

Altogether, our data indicate that miR126 and miR145 are regulated within the released EVs during the development of the fibrotic damage during diabetes occurrence, supporting their possible role in further patients’ stratification. In addition, we link the presence of those miRNAs within uEVs with their release by podocytes and tubular epithelial cells under diabetes-mimicking conditions.

## 4. Materials and Methods

### 4.1. Study Groups

All patients enrolled in the present study provided informed written consent for the study. The study protocol was approved by the Bioethics Committee of the A.O.U. Città della Salute e della Scienza Hospital (protocol no. 0021671). The study was conducted according to the principles expressed by the Declaration of Helsinki of 1975, as revised in 2013. The study group was composed of a total of 46 adult patients with type 2 diabetes admitted to the clinic (HbA1c > 48 mmol/mol) and divided based on their albuminuria levels in normoalbuminuria, microalbuminuria and macroalbuminuria patients.

### 4.2. Patients’ Urine Collection and uEV Isolation

Morning urine samples (~100 mL) were collected in sterile containers. In parallel, biochemical analysis was performed by the clinical laboratory of the A.O.U. Città della Salute e della Scienza Hospital. Urine samples were centrifuged at 3000 rpm for 15 min to remove whole cells, large membrane fragments and other debris. Protease Inhibitor (PI) Cocktail (Sigma-Aldrich. St. Louis, MO, USA, 100 μL PI/100 mL urine) and NaN_3_ (10 mM, Sigma-Aldrich) were added immediately to the remaining supernatant. After filtration through 0.8- and 0.45-µm filters (Merck Millipore, Burlington, MA). uEVs were collected from the samples through ultracentrifugation (Beckman Coulter, OPTIMA L-100 K Ultracentrifuge, Rotor Type 70-Ti, Brea, CA, USA) at 100,000× *g* for 1 h at 4 °C, as described [21]. The pellet was then resuspended in RPMI (Euroclone, Turin, Italy) + 1% DMSO (Sigma-Aldrich) and stored at −80 °C until use.

### 4.3. Cell Lines

HK2 cells (ATCC) were cultured in Dulbecco’s Modified Eagle Medium Low Glucose (DMEM LG) supplemented with 10% foetal bovine serum (FBS) (Euroclone S.p.A., Pero, MI, Italy). Immortalized human podocytes, previously characterized [22] were cultured in DMEM High Glucose (Euroclone) with 10% FBS. Glomerular endothelial cells (GEC) (Cell Biologics Inc., Chicago, IL, USA) previously immortalized [22] were cultured in EndoGRO-LS Complete Culture Media (Merck Millipore) and 10% FBS. Penicillin-Streptomycin (PS) and L-glutamine were added to the cell cultures.

### 4.4. In Vitro Diabetes-Mimicking Models

For the glomerular model of diabetic induction, a millifluidic device (IVtech Srl., Lucca, Italy) was used as previously described [22]. The model was assembled in a structure in which GEC and podocytes were separated with a layer of collagen IV (Sigma-Aldrich). Diabetic induction was achieved by adding Transforming Growth Factor-β1 (TGF-β1) (Sigma-Aldrich, 30 ng/mL) in EV-deprived complete culture medium for 24 h. For the tubular model, HK2 were treated with human serum albumin (HAS, 10 mg/mL) and high glucose (HG, 30 mM) for 72 h in RPMI 1640 (Gibco BRL, Paisley, UK). Control cells remained in basal DMEM LG.

### 4.5. EV Isolation from Cell Supernatant

EVs were collected from the supernatant of kidney cells after 24 h treatment (in the presence of complete medium deprived of EVs by ultracentrifugation). Following the removal of cell debris and apoptotic bodies by centrifugation at 3000× *g* for 15 min and microfiltration over a 0.22-μm filter. Medium was then centrifuged at 4000 rpm for 15 min with the 3 KDa Amicon filters (Amicon-Merck Millipore). EVs were then purified using the ExoQuick (SystemBio, Palo Alto, CA, USA) following the manufacturer’s protocol. Briefly, ExoQuick was added to the EV containing supernatant and incubated for 2 h at 4 °C. The mixture was then centrifuged at 1500× *g* for 30 min to separate supernatant and the EV-enriched pellet.

### 4.6. Transmission Electron Microscopy

Transmission electron microscopy (TEM) was performed on EVs placed on 200-mesh nickel formvar carbon-coated grids (Electron Microscopy Science) for 20 min to promote adhesion. The grids were then incubated with 2.5% glutaraldehyde plus 2% sucrose. EVs were negatively stained with NanoVan (Nanoprobes, Yaphank, NY, USA) and observed using a Jeol JEM 1400 Flash electron microscope (Jeol, Tokyo, Japan) [37]. 

### 4.7. Nanoparticle Tracking Analysis

EVs were analyzed using nanoparticle tracking analysis using the NanoSight NS300 system (NanoSight, Salisbury, UK) configured with a blue 488-nm laser and a high-sensitivity digital camera system (OrcaFlash 2.8, Hamamatsu C1 1440, NanoSight). Briefly, EVs stored in −80 °C were thawed, strongly vortexed and properly diluted in saline solution (Fresenius Kabi, Bad Homburg, Germany) previously filtered with a 0.1-µm filter (Merck Millipore). For each sample, three videos of 30 s each were recorded. The settings of acquisition and analysis were optimized and kept constant between samples.

### 4.8. Super-Resolution Microscopy 

A super-resolution experiment was performed using a Nanoimager S Mark II microscope from ONI (Oxford Nanoimaging, Oxford, UK) equipped with a 100×, 1.4 NA oil immersion objective, an XYZ closed-loop piezo 736 stage and triple emission channels split at 488, 555 and 640 nm. For the evaluation of tetraspanin expression, the EV MAN profiler Kit (ONI) was used following the manufacturer’s protocol. Fluorescent antibodies anti CD9-488, CD63-568 and CD81-647 were included in the kit. For the evaluation of renal markers, 2.5 µg of purified rabbit anti-Aquaporin (AQP) 1 and AQP2 (Santa Cruz) were conjugated with Alexa Fluor 647 dye, using the Apex Antibody Labelling Kit (Invitrogen, Carlsbad, CA, USA) according to the manufacturer’s protocol. AQP1 and AQP2 were used in the EV MAN profiler Kit in a combination of CD63-568 present in the kit. The samples were washed twice with PBS and 10 μL of ONI Bcubed Imaging Buffer was added for acquisition. Three or two-channel dSTORM data were acquired sequentially at 30 Hz (Hertz) in total reflection fluorescence (TIRF) mode. Single molecule data was filtered using NimOS (Version (v.1.18.3, ONI) based on the point spread function shape, photon count and localization precision to minimize background noise and remove low-precision localizations [38].

### 4.9. MACSPlex Exosome Kit

EV surface markers were evaluated by cytofluorimetric analysis using the MACSplex Exosome Kit (Miltenyi). A total number of 1 × 10^9^ EVs were used following the manufacturer’s protocol [24]. The acquisition was performed with a BD FACS Celesta Cell analyzer (BD, Franklin Lakes, NJ, USA). The evaluation of the exosome marker levels was calculated first by subtracting the background fluorescence and then normalizing the median fluorescence of each marker with the mean of the median fluorescence of the tetraspanin markers (CD63-CD81-CD9). Acquisition and analysis settings were optimized and kept constant between samples.

### 4.10. Total RNA Extraction

Total RNA, both from patients’ urine- and cell-derived EVs, was extracted using the mirVana kit (Thermo Fisher Scientific. Waltham, MA, USA) as per the manufacturer’s protocol. Isolated RNA was quantified using the NanoDrop2000 spectrophotometer (Thermo Fisher Scientific) and either used immediately or stored at −80 °C until further use.

### 4.11. RNA and miRNA Analysis

For the gene expression analysis in kidney cells, first-strand cDNA was produced from 200 ng of total RNA using the High-Capacity cDNA Reverse Transcription Kit (Thermo Fisher Scientific). Quantitative Real-time (qRT)-PCR experiments were performed in the 20-μL reaction mixture containing 5 ng of cDNA template, the sequence-specific oligonucleotide primers (purchased from MWG-Biotech. Table 2) and the Power SYBR Green PCR Master Mix (Thermo Fisher Scientific). GAPDH mRNA was used to normalize the RNA inputs.

EV miRNA analysis was carried using the miRCURY LNA™ Universal RT microRNA PCR kit (Qiagen, Düsseldorf, Germany). In selected experiments, the spike-in UNISP6 was added during the RNA isolation as additional extraction control. Fifty pg of reverse transcription reaction products were combined with SYBR Green Master Mix (Qiagen) and LNA™ PCR primer mix and analyzed as described by the manufacturer’s protocol. RNU6B or UNISP6 were used as qRT-PCR loading controls based on their stability in the different conditions tested.

### 4.12. Statistical Analysis

Statistical analyses were performed using Graph Pad Prism version 5.04 (Graph Pad Software Inc., La Jolla, CA, USA). Comparison between groups were either analyzed using non-parametric (Mann–Whitney) or parametric (Student’s *t*-test and one-way ANOVA with Dunnett’s multiple comparison test), when appropriate. A *p* value of <0.05 was considered significant.

## 5. Conclusions

In conclusion, the changes of miRNA expression in uEVs can be a useful tool for the diagnosis of the different stages of DN. The present study first validates the increase in miR126 and miR145 in uEVs in DN and, more importantly, correlates their increase with the progression of DN, as assessed by proteinuria levels. Moreover, we here were confirmed their altered release in EVs derived from renal cells undergoing diabetes-mimicking damage. It can be speculated that the simultaneous evaluation of the two biomarkers may increase the sensitivity of detecting the renal damage progression in this pathology.

## Figures and Tables

**Figure 1 ijms-23-12098-f001:**
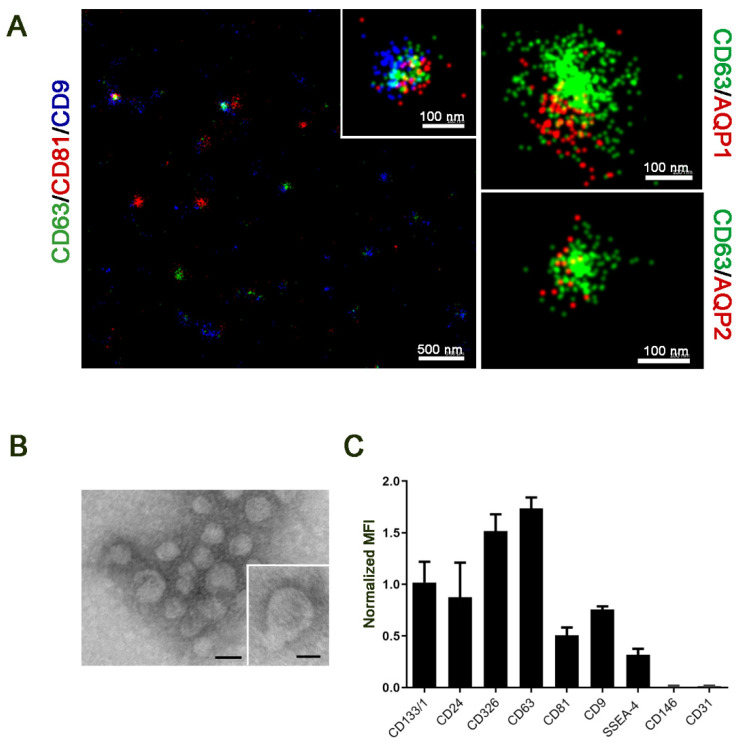
Urinary EVs characterization. (**A**) Representative super resolution microscopy images of DN uEVs showing a heterogeneous distribution of tetraspanins (left panel, CD63 in green, CD81 in red and CD9 in blue); in the inset a single uEV expressing the three tetraspanins. The coexpression of AQP1 and AQP2 (in red) with CD63 (in green) (right panels) was observed. The scale bars are below each EV image (500 and 100 nm, respectively). (**B**) Representative micrograph of TEM of uEVs (Scale bar: 100 nm, inset: 50 nm). (**C**) MACSPlex bead-based flow cytometry analysis for different surface markers in uEVs. The relative marker levels were normalized to the mean of the median tetraspanin intensity (normalized MFI) (data are mean of n = 3 uEV preparations).

**Figure 2 ijms-23-12098-f002:**
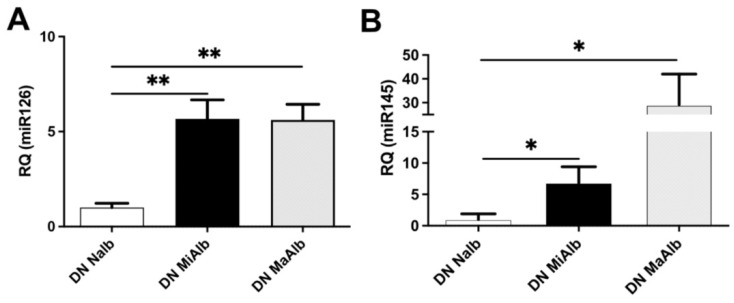
Relative quantification of miR126 (**A**) and miR145 (**B**) in the uEVs of patients divided in normoalbuminuric (NAlb), microalbuminuric (MiAlb) and macroalbuminuric (MaAlb) patients and normalized to RNU6B (* *p* < 0.05 and ** *p* < 0.001).

**Figure 3 ijms-23-12098-f003:**
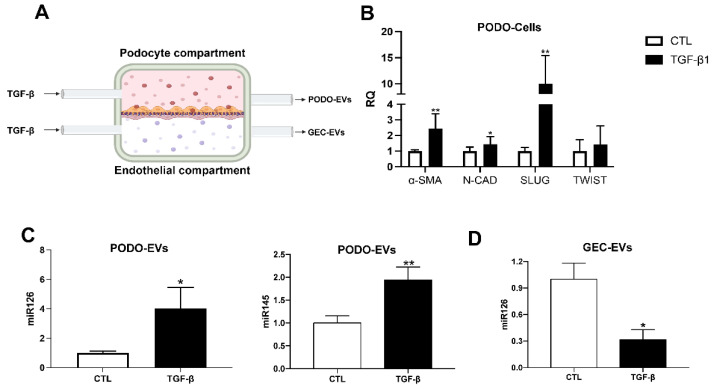
Schematic image of the millifluidic dynamic glomerular model (**A**) The podocyte and endothelial compartments were separated by a membrane coated with collagen type IV. TGF-β was introduced into the circuit through the inlet channels, while the outgoing medium, enriched with EVs, was collected separately. (**B**) Relative quantification of EMT genes (α-SMA, N-CAD, SLUG and TWIST) in podocytes (PODO-Cells) cultured in the glomerular model and treated with TGF-β. Untreated podocytes were used as control (CTL) (* *p* < 0.05: ** *p* < 0.01 vs. CTL). (**C**,**D**) Relative quantification of miR126 and miR145 in EVs (**C**) from TGF-β and CTL podocytes (PODO-EVs) and (**D**) glomerular endothelial cells (GEC-EVs) (n = 4 independent experiments) (* *p* < 0.05: ** *p* < 0.01 vs. CTL).

**Figure 4 ijms-23-12098-f004:**
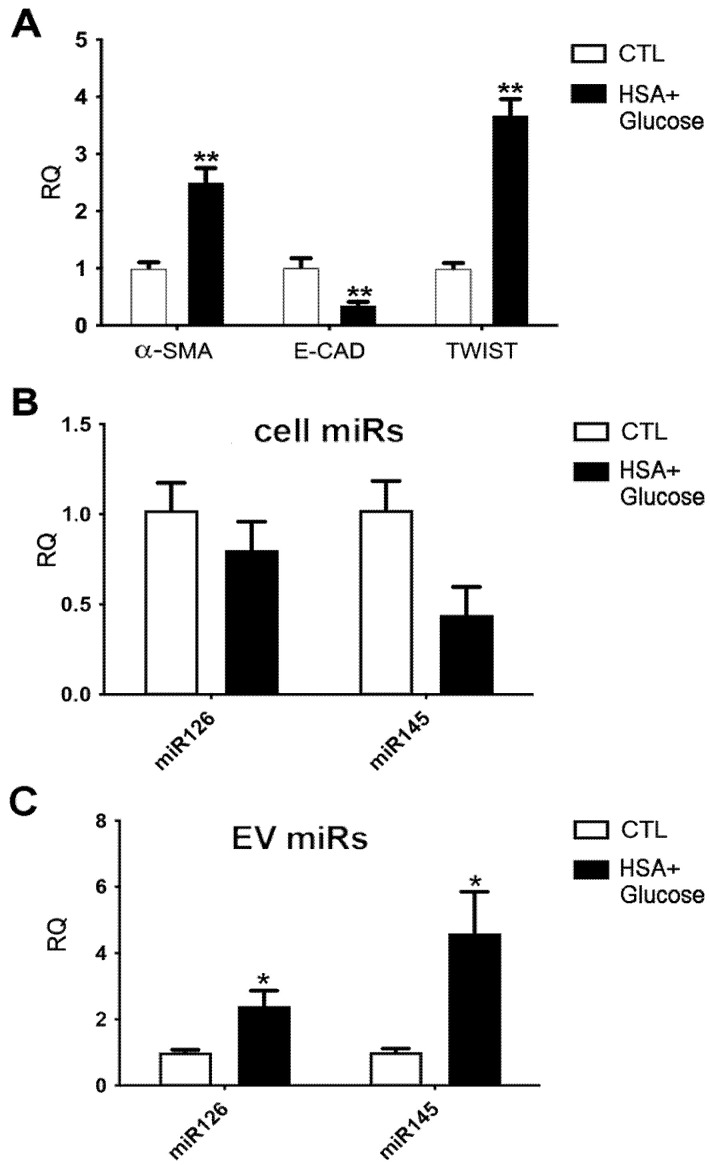
(**A**) Relative quantification of EMT genes α-SMA, E-cadherin and TWIST in human proximal tubular cells (HK2) treated with 10 mg/mL of HSA and 30 mM Glucose. (**B**, **C**) MiR126 and miR145 levels were evaluated in HK2 in diabetic-like conditions (**B**) and in their released EVs (**C**) Untreated HK2 and their released EVs were used as control (CTL) (n = three independent experiments). (* *p* < 0.05: ** *p* < 0.001 vs. CTL).

**Table 1 ijms-23-12098-t001:** Clinical characteristics of diabetic patients.

Patient Group	N.	Sex (Male/Female)	Age (Years)	Creatinine (mg/dL)	eGFR (mL/min)	Albuminuria (mg/dL)	HbAlc. (mmol/mol)	EV Mean and Mode Size (nm)
NAlb DN	17	11/6	68 (63–77)	1.16 ± 0.12	72.4± 6.6	0.1 ± 0.1	74.1 ± 6.2	Mean: 264.9 ± 8.2 Mode: 207.7 ± 8.6
MiAlb DN	15	10/5	79 (72–85)	1.40 ± 0.24 a	62.4± 6.4	22.4 ± 2.2 c	58.7 ± 4.4	Mean: 251.1 ± 3.2 Mode: 198.3 ± 7.5
MaAlb DN	14	8/6	82 (70–87)	1.57 ± 0.22 a	45.1± 7.1 b	129.3 ± 27.2 d	58.8 ± 3.1	Mean: 250.2 ± 6.7 Mode: 195.9 ± 9.9

Data are reported as means ± SE. Age is reported as median (Q1–Q3). NAlb DN: patients with diabetic normoalbuminuria; MiAlb DN: patients with diabetic microalbuminuria; MaAlb DN: patients with diabetic macroalbuminuria. (a) Creatinine vs. patients with NAlb DN (*t*-test *p* < 0.05); (b) Estimated glomerular filtration rate vs. patients with NAlb DN (Mann–Whitney test *p* = 0.007); (c) albuminuria vs. patients with NAlb DN (Mann–Whitney test *p* = 2.25 × 10^−7^); (d) albuminuria vs. patients with NAlb DN (Mann–Whitney test *p* = 1.44 × 10^−7^) and vs. patients with MiAlb DN (Mann–Whitney test *p* = 1.18 × 10^−6^). EV size distribution is expressed both as mean (nm) and mode of the vesicles diameter (nm).

**Table 2 ijms-23-12098-t002:** Oligonucleotide primers sequences for qRT-PCR experiments.

Gene	Forward Sequence	Reverse Sequence
*GAPDH*	F-GTCTCCTCTGACTTCAACAGCG	R-ACCACCCTGTTGCTGTAGCCAA
*N-cadherin*	F-CCTCCAGAGTTTACTGCCATGAC	R-GTAGGATCTCCGCCACTGATTC
*E-cadherin*	F-GCCTCCTGAAAAGAGAGTGGAAG	R-TGGCAGTGTCTCTCCAAATCCG
*Twist family bHLH transcription factor 1(TWIST1)*	F-GCCAGGTACATCGACTTCCTCT	R-TCCATCCTCCAGACCGAGAAGG
*α-smooth muscle cell actin* *(α-SMA)*	F-CTATGCCTCTGGACGCACAACT	R-CAGATCCAGACGCATGATGGCA
*Snail family transcriptional* *repressor 2 (SLUG)*	F-AAACTACAGCGAACTGGACACACA	R-GAGCAGCGGTAGTCCACACAG
*RNU6B*	F-TGCGGCTGCGCAAGGATGA	
*miR126*	F-CATTATTACTTTTGGTACGC	
*miR145*	F-TGCGGTGAGATGAAGCACT	

## Data Availability

Not applicable.

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
