# Peer review of "Upregulation of miR145 and miR126 in EVs from Renal Cells Undergoing EMT and Urine of Diabetic Nephropathy Patients"

_ijms, 2022, doi:10.3390/ijms232012098_

Round 1
Reviewer 1 Report
The primary problem with the manuscript is that the authors did not characterize EVs correctly. Although the authors write in the methodology that they used NTA, it is not a sufficient method to characterize EVs properly. The authors should use methods that allow them to describe surface marker characteristics for a specific EV pool and provide their mean size (this is what NTA is used for).
Author Response
Reviewer 1
We thank the reviewer for the careful evaluation of our manuscript, and we revised it accordingly.
Q1. The primary problem with the manuscript is that the authors did not characterize EVs correctly. Although the authors write in the methodology that they used NTA, it is not a sufficient method to characterize EVs properly. The authors should use methods that allow them to describe surface marker characteristics for a specific EV pool and provide their mean size (this is what NTA is used for).
R1. We agree with the reviewer that NTA is not a sufficient method for EV characterization, and we revised the manuscript to add the data on uEV characterization. In detail, we added a deep characterization of uEVs isolated from diabetic patients using super resolution microscopy, transmission electron microscopy and MACSplex cytofluorimetric analysis
In Figure 1, we now report the presence in uEVs of the classical tetraspanins (CD63, CD9 and CD81) by both super resolution microscopy at single EV level and by bead-based cytofluorimetric analysis, that also shows the presence of the urinary EV markers CD24, CD133 and CD326; and we show the expression of the AQP1 and AQP2, nephron segments’ markers, by super resolution microscopy. Morphology, integrity and standard size distribution was also confirmed in uEVs by super resolution and TEM. Moreover, mean and median size distribution of uEVs as evaluated by NTA was added in Table 1.
Please see Methods: page 4 and 5, lines 158-163 and 174-199, Results: page 6 lines 233-243 and the new Figure 1.
Reviewer 2 Report
The manuscript by Dimuccio et al investigates miR145 and miR126 in EVs from renal cells undergoing EMT and urine of diabetic nephropathy patients. The work may be potentially interesting, but there is a lot of information missing, and also some controls. Although some references are provided, it is of key relevance to explain to readers the reasons underlining the choice to investigate these two specific miRNAs, as it is unclear what could be the reason underlining their expression in ND. Besides, some key data are lacking. Authors did not report the number of uEVs in normal and diabetic patients and did not assay additional miRNAs that are possibly not altered in diabetic patients and cell models, so it is unclear whether the altered expression of miR126 and miR146 is specific or there is a generic increase in miRs content in uEVs of diabetic patients.
Major points
In the Simple Summary, the link between “the molecular mechanism associated with the glomerular and tubular damage” and “epithelial to mesenchymal transition” should be more explicit
In the Abstract, the rationale of the focus on miR126 and miR145 is unclear, and the same in the Introduction
In the Results section, briefly introduce how uEVs were separated and characterized
In Figure 2, the expression of miR126 and miR145 in uEVs is reported, but it is not specified if these are expressed in cells: is it the lack of the detection of miR145 in glomerular endothelial cells a lack of expression or a lack of packaging into EVs? How was the normalization strategy chosen for miRs qRT-PCR, as RNAs used for normalization of cell miRNA may be not suitable for EVs
Please avoid sentences such as “…and an increasing shift in miR126 level, although it did not reach significance (Figure 3B).” If the increase was not statistically significant, it cannot be defined an increase, but it is just a fluctuation
What is known about miR126 and miR145 biology? Where are they located and how are they transcribed? What is known about their function? What could be molecular reason underlining their abnormal regulation in ND?
Author Response
Reviewer 2
We thank the reviewer for the detailed evaluation of our manuscript, and we did our best to comply with the requests.
The manuscript by Dimuccio et al investigates miR145 and miR126 in EVs from renal cells undergoing EMT and urine of diabetic nephropathy patients. The work may be potentially interesting, but there is a lot of information missing, and also some controls. Although some references are provided, it is of key relevance to explain to readers the reasons underlining the choice to investigate these two specific miRNAs, as it is unclear what could be the reason underlining their expression in ND. Besides, some key data are lacking. Authors did not report the number of uEVs in normal and diabetic patients and did not assay additional miRNAs that are possibly not altered in diabetic patients and cell models, so it is unclear whether the altered expression of miR126 and miR146 is specific or there is a generic increase in miRs content in uEVs of diabetic patients.
We considered all the Reviewer’ comments and revised the manuscript accordingly.
In details:
In the previous version, we did not present the preliminary experiments leading to the choice of miR145 and miR126. We agree that this aspect is of importance to better understand the rational of our work. We now report that the miRNAs investigated in this work were initially selected among a list of miRNAs already known to be directly or indirectly involved in diabetes or in tissue fibrosis. This is better detailed in the answer to Question 2, below.
The number of EVs measured by NTA can be questioned based on the limitation of the method in determining EV concentration as pointed out also by the reviewers 1 and 3, therefore we removed this information and add the EV mean size and mode measurement performed by NTA.
Major points:
Q1: In the Simple Summary, the link between “the molecular mechanism associated with the glomerular and tubular damage” and “epithelial to mesenchymal transition” should be more explicit.
R1. In the Simple Summary, we added a phrase to make clearer the link between renal cell damage and EMT. In particular, we now report that “MiR145 and miR126 increased in parallel in EVs from renal epithelial cells undergoing diabetes-like damage and transition toward a fibrotic mesenchymal phenotype”.
Q2: In the Abstract, the rationale of the focus on miR126 and miR145 is unclear, and the same in the Introduction.
R2. We modified the Abstract and the Introduction adding a clearer explanation of the rationale behind our study and the miRNA selection. We now better explain that the miRNAs investigated in this work were initially selected among a list of miRNAs already known to be directly or indirectly involved in diabetes or in tissue fibrosis, as miR21, miR24, miR296, miR320c, miR221, miR126 and miR145 (refs 12-20). qRT-PCR evaluation of these miRNA group is reported in the new Supplementary Table 1.
Among the miRNA evaluated, only miR126 and miR145 showed significant differences among DN groups and were further evaluated in a larger patient cohort. Our data showed that the altered expression of miR126 and miR145 in diabetic patients is specific in respect to other miRNAs that were not modified in DN progression.
See Abstract, lines 30-31, Introduction: page 2, lines 80-84, Results: page 7, lines 259-263, Discussion: lines 331-333 and Supplementary Tables 1 and 2.
Q3. In the Results section, briefly introduce how uEVs were separated and characterized.
R3. The reference associated to the method used for EVs separation (21) was added in the Result section. Moreover, we added new data reporting a deep characterization of uEVs isolated from diabetic patients using super resolution microscopy, transmission electron microscopy and MACSplex cytofluorimetric analysis. In particular, in Figure 1, we now report the presence in uEVs of the classical tetraspanins (CD63, CD9 and CD81) by both super resolution microscopy at single EV level and by bead-based cytofluorimetric analysis, that also shows the presence of the urinary EV markers CD24, CD133 and CD326; and we show the expression of the AQP1 and AQP2, nephron segments’ markers, by super resolution microscopy. Morphology, integrity and standard size distribution was also confirmed in uEVs by super resolution and TEM. Moreover, mean and median size distribution of uEVs as evaluated by NTA was added in Table 1. Please see Methods: page 4 and 5, lines 158-163 and 174-199, Results: page 6 lines 233-243 and the new Figure 1.
Q4. In Figure 2, the expression of miR126 and miR145 in uEVs is reported, but it is not specified if these are expressed in cells: is it the lack of the detection of miR145 in glomerular endothelial cells a lack of expression or a lack of packaging into EVs?
R4. We agree with the reviewer that this is indeed an interesting point. We performed new experiments of miR126 and miR145 modulation in renal cells in the in vitro experiments. In details, we showed that both miRNAs are present in glomerular and tubular cells. MiR126 and miR145 levels did not change in treated tubular cells, nor in endothelial cells. In podocytes, diabetes-mimicking conditions only induced miR145 increase in podocytes. Data on glomerular cells are reported in Results: page 8, lines 284-286 and 291-292 and Suppl. Fig.1, data on tubular cells are reported on page 9, lines 310-311 and Fig. 4B).
We also discussed that pathological stimuli such as TGF-beta as well as albumin and glucose overload mat directly affect the package of miRNA126 in EVs from renal cells (see Discussion: page 11, lines 349-353).
Q5. How was the normalization strategy chosen for miRs qRT-PCR, as RNAs used for normalization of cell miRNA may be not suitable for EVs. Please avoid sentences such as “…and an increasing shift in miR126 level, although it did not reach significance (Figure 3B).” If the increase was not statistically significant, it cannot be defined an increase, but it is just a fluctuation
R5. The normalization strategy was based on the use of the RNU6B or UNISP6 as loading controls based on their stability in the different conditions tested. The RNU6B expression was not modified in treated kidney cells, showing its stability in our setting. For EV-miRNA screening, UNISP6 was added to the analysis and used as loading control to eliminate possible interferences associated with the snoRNAs modulation in cell derived EVs. This information was added to Results: page 5, lines 217-223.
We acknowledge the comment of Reviewer 2 on the significance of miRNA expression by qRT-PCR. We performed ad additional experiment in human proximal tubular cells (HK2) treated with albumin and glucose overload, that allowed to obtain statistical significance. See new Figure 4C.
Q6. What is known about miR126 and miR145 biology? Where are they located and how are they transcribed? What is known about their function? What could be molecular reason underlining their abnormal regulation in ND?
R6. We added a deep evaluation of the literature data supporting the possible role of miR145 and miR126 in EMT and in fibrosis (Please see Discussion Page 10, lines 353-357 and 365-371). In particular, the role of miR126 in EMT is supported by previous studies showing its role in the development of liver fibrosis in hepatic stellate cells [29] and, in systemic sclerosis, in collagen modulation [30]. At variance, the role of miRNA145 in renal tissue damage is unclear, as it can positively or negatively regulate fibrosis in different pathological processes [31-34]. In tumor cells, miR145 prevented cell invasion and EMT [33]. At variance, in peritoneal fibroblasts, TGF-beta induced miR145 accounted for EMT induction and fibrosis development through FGF10 decrease [31]. Similarly, in podocytes, miR145 enhancement was recently correlated with foot process effacement and the development of proteinuria [34].
We also discuss the limitation of the present study. Beside the role as biomarkers, the possible functional effect of EV-contained miR126 and miR145 would be of interest, considering that both miRNAs may affect positively and negatively EMT [28-36]. It can be speculated that uEV cargo during DN may be involved in the amplification of diabetic-induced alterations via EVs release or, alternatively, simply represent the modulation of the miRNA packaging in EVs from damaged renal cells. Further experiments are needed to clarify this aspect. See Discussion, page 11, lines 375-380.
Reviewer 3 Report
Dimuccio et al identifed renal cell-derived extracellular vesicle (EV)-associated miRNA-145 and miRNA-126 as urine biomarkers of kidney damage in patients with type-2 diabetes. There are several issues that should be addressed before considering this manuscript for publication in Int J Mol Sci.
Comments:
- The novelty of this study is hampered by the fact that miRNA-126 (PMID:29981742, 29689545) and miRNA-145 (PMID: 24223694) were already described as urinary markers of diabetic nephropathy. Authors should highlight how their manuscript advances in the field.
- Why authors choose to study these two miRNAs (miR-126 and miR-145) and not others? Please explain the rationale.
- Table 1 - MaAlb DN group: It is stated that sample size is n=15 but then only 8 men and 6 women are reported. Are mean values of the other parameters of this group based on n=14 or n=15? Please revise.
- How authors explain the fact that patients with diabetic microalbuminuria have less EV levels than patients with diabetic normoalbuminuria?
- Nanoparticle tracking analysis is a method for size determination rather than quantification.
- For low yield (such as RNA from EVs), RNA quantification by Qubit is recommended instead of Nanodrop.
- Figure 1 - It should be specified how was normalised the relative quantification of urinary miRNAs.
- Number of independent experiments should be stated for each cell culture study (e.g. in the figure legend).
- The molecular mechanisms by which miRNA-126 and miRNA-145 intervene in the renal damage associated with type-2 diabetes is not investigated. Mechanistic studies are required.
Author Response
Reviewer 3
We thank the reviewer for the careful evaluation of the manuscript and for the suggestions, that in our opinion improved the quality of the study.
Q1. The novelty of this study is hampered by the fact that miRNA-126 (PMID:29981742, 29689545) and miRNA-145 (PMID: 24223694) were already described as urinary markers of diabetic nephropathy. Authors should highlight how their manuscript advances in the field.
R1. We agree with the comment and changed the text to bring more relevance to the aspects associated with the novelty of the data in our manuscript. MiR126 increase was previously described in uEVs in generic diabetic condition, and miR145 increase in uEV from microalbuminuric patients. Our data validated in a different cohort the increase in miR126 and miR145 in uEVs observed by other studies in diabetic conditions. This is of importance in the context of marker validation. More importantly, we for the first time correlate their increase with progression of DN, as assessed by albuminuria levels, supporting their possible role in further patients’ stratification.
Moreover, no data on miR126 and miR145 release by different kidney cells submitted to a profibrotic stimulation were previously assessed. In this context, our results for the first time were able to directly correlate presence of those miRNAs within uEVs with their release by podocytes and tubular epithelial cells under diabetes-mimicking conditions. Finally, we added new experiments showing that miR126 in particular was not modulated in cells, but rather in EVs, after renal cell EMT, suggesting a specific effect on EV packaging.
See Discussion, page 11, lines 359-362 and 381-385 and page 12, lines 387-393.
Q2. Why authors choose to study these two miRNAs (miR-126 and miR-145) and not others? Please explain the rationale.
R2. We thank the reviewer for the important comment. In the previous version, we did not present the preliminary experiments leading to the choice of miR145 and miR126. We agree that this aspect is of importance to better understand the rational of our work. As we already reported in the response to Reviewer 2, we now better explain that the miRNAs investigated in this work were initially selected among a list of miRNAs already known to be directly or indirectly involved in diabetes or in tissue fibrosis, as miR21, miR24, miR296, miR320c, miR221, miR126 and miR145 (refs 12-20). qRT-PCR evaluation of these miRNA group is reported in the new Supplementary Table 1.
Among the miRNA evaluated, only miR126 and miR145 showed significant differences among DN groups and were further evaluated in a larger patient cohort. Our data showed that the altered expression of miR126 and miR145 in diabetic patients is specific in respect to other miRNAs that were not modified in DN progression.
See Abstract, lines 30-31, Introduction: page 2, lines 80-84, Results: page 7, lines 259-263, Discussion: lines 331-333 and Supplementary Tables 1 and 2.
Q3. Table 1 - MaAlb DN group: It is stated that sample size is n=15 but then only 8 men and 6 women are reported. Are mean values of the other parameters of this group based on n=14 or n=15? Please revise.
R3. We thank you the reviewer for pointing out this typo in Table 1, that was corrected accordingly.
Q4. How authors explain the fact that patients with diabetic microalbuminuria have less EV levels than patients with diabetic normoalbuminuria? Nanoparticle tracking analysis is a method for size determination rather than quantification.
R4. We agree that the number of EVs measured by NTA can be questioned based on the limitation of the method in determining EV concentration, as also pointed out by Reviewer 1. In agreement with these comments, we removed this information and we added the EV mean size and mode measurement performed by NTA in Table 1.
Q5. For low yield (such as RNA from EVs), RNA quantification by Qubit is recommended instead of Nanodrop.
R5. We agree with the comment of Reviewer 3. Because we did not have the possibility to use fluorescence-based detection of RNA isolated from EVs, we alternatively added during the RNA isolation the spike-in UNISP6 as extraction control and tested its expression during the qRT-PCR experiments. It is now specified in the Methods, see page 5, lines 217-218.
Q6. Figure 1 - It should be specified how was normalized the relative quantification of urinary miRNAs.
R6. We added the normalization method in the legend of the new Figure 2.
Q7. Number of independent experiments should be stated for each cell culture study (e.g. in the figure legend).
R7. We thank the Reviewer, these data were added to the legends of the new Figure 3 and 4.
Q8. The molecular mechanisms by which miRNA-126 and miRNA-145 intervene in the renal damage associated with type-2 diabetes is not investigated. Mechanistic studies are required.
R8. We agree that limitation in our study is the absence of the definition of the molecular mechanism associated with the regulation of miR126 and 145 and their activity in damaged kidney cells. We added the following phrase: Beside the role as biomarkers, the possible functional effect of EV-contained miR126 and miR145 would be of interest, considering that both miRNAs may affect positively and negatively EMT [28-36]. It can be speculated that uEV cargo during DN may be involved in the amplification of diabetic-induced alterations via EVs release or, alternatively, simply represent the modulation of the miRNA packaging in EVs from damaged renal cells. Further experiments are needed to clarify this aspect. See Discussion, page 11, lines 375-380.
However, a specific mechanistic evaluation of miR function is partially outward the scope of the manuscript. We here focused on the identification of the kidney cells potentially responsible for the uEV-associated miR release and the biological process behind this phenomenon. In this context, our results for the first time were able to directly correlate these miRNAs with EMT and fibrosis in the kidney, showing the enrichment in their package in EVs selectively released by damaged podocytes and tubular epithelial cells.
We also now added a better description of the literature data supporting the possible role of miR145 and miR126 in EMT and in fibrosis (Please see Discussion Page 10, lines 353-357 and 365-371). In particular, the role of miR126 in EMT is supported by previous studies showing its role in the development of liver fibrosis in hepatic stellate cells [29] and, in systemic sclerosis, in collagen modulation [30]. At variance, the role of miRNA145 in renal tissue damage is unclear, as it can positively or negatively regulate fibrosis in different pathological processes [31-34]. In tumor cells, miR145 prevented cell invasion and EMT [33]. At variance, in peritoneal fibroblasts, TGF-beta induced miR145 accounted for EMT induction and fibrosis development through FGF10 decrease [31]. Similarly, in podocytes, miR145 enhancement was recently correlated with foot process effacement and the development of proteinuria [34].
Round 2
Reviewer 1 Report
The authors significantly improved the manuscript. However, there are still issues to explain:
1. Table 1 presents the results. It should therefore be moved to the appropriate subsection of the manuscript.
2. What does EV mode size mean (table 1)?
3. There is no statistical comparison between the subgroups in table 1 - did age and other characteristics differ between the subgroups?
4. The authors need to discuss the potential miR-mRNA and/or ncRNA-miR-mRNA axis in the context of the nucleic acids tested.
5. References must be extended with the latest publications:
https://www.mdpi.com/2073-4409/11/18/2913/htm
https://www.mdpi.com/1422-0067/22/17/9586
Author Response
We thank the reviewer for the careful evaluation of our manuscript, and we revised it accordingly.
The authors significantly improved the manuscript. However, there are still issues to explain:
- Table 1 presents the results. It should therefore be moved to the appropriate subsection of the manuscript.
Response: As suggested, we moved the Table (now Table 2) to the Results section, page 7.
- What does EV mode size mean (table 1)?
Response: We calculated both the mean diameter and mode diameter distribution of urinary EVs. The NTA software calculated the mode distribution as the most frequently occurring diameter value calculated in a population of heterogeneous vesicles.
We better explained in the Table legend that EV size distribution is expressed both as mean (nm) and mode of the vesicles diameter (nm).
- There is no statistical comparison between the subgroups in table 1 - did age and other characteristics differ between the subgroups?
We now better describe the statistical comparison between subgroups in Table 1 (now 2), lines 242-247. A significant difference in creatinine, glomerular filtration rate and albuminuria but not in age or glycated hemoglobin values was observed.
- The authors need to discuss the potential miR-mRNA and/or ncRNA-miR-mRNA axis in the context of the nucleic acids tested.
- References must be extended with the latest publications:
https://www.mdpi.com/2073-4409/11/18/2913/htm
https://www.mdpi.com/1422-0067/22/17/9586
We added that Understanding these intracellular mechanisms and precisely following the axis of miRNA-messenger RNA in kidney cells [25, 26] is crucial for the future use of EVs in the clinical evaluation of DN, citing the indicated references.
Reviewer 2 Report
The manuscript by Dimuccio et al has greatly improved the information related to EVs isolation and properties, providing new information on EVs, and clearly specifying the main characteristic related to the patients enrolled and the rationale underlining the choice of the two miRNAs. Nevertheless, a few points are still unclear.
Main points
In their response, authors mentioned that NTA data are not useful and there is no count of EVs reported. Could authors explain why? What confounding factors can affect this data? Is it a technical limit or it is related to the matrix (urine)? As for examples the reason of this unusefulness is the interference by different abundance of proteins or there are other reasons? This should be explained to readers, even because some protocols (MACSPlex exosome kit) rely on EV count.
Line 263 The sentence “In preliminary experiments, no significant modulation of miR21, miR24, miR221, miR296, and miR320c 264 was observed (Suppl. Table 1).” Apparently describe a PCR experiment, but actually in the supplementary material there is just a table with oligo used. The sentence and the Supplementary material should be coherent
The sentence “At variance, the level of miR145 and miR126 in uEVs from diabetic patients appeared significantly modulated within the groups.” Is unclear. What do authors mean for “At variance”?
In Figure 3B, the indication of PODO-Cells would help to catch its meaning
Line 286 Please correct the sentence “In the millifluid system, podocytes after TGF-b stimulation showed increased in the expression…”
Line 293 Authors wrote that “No significant changes in miR126 or miR145 levels were observed in endothelial cells when treated with TGF-b(not shown).” Could authors provide these data in the Supplementary material?
Author Response
We thank the reviewer for the detailed evaluation of our manuscript, and for the suggestions that we think improved the clarity of the data presentation.
The manuscript by Dimuccio et al has greatly improved the information related to EVs isolation and properties, providing new information on EVs, and clearly specifying the main characteristic related to the patients enrolled and the rationale underlining the choice of the two miRNAs. Nevertheless, a few points are still unclear.
Main points
- In their response, authors mentioned that NTA data are not useful and there is no count of EVs reported. Could authors explain why? What confounding factors can affect this data? Is it a technical limit or it is related to the matrix (urine)? As for examples the reason of this unusefulness is the interference by different abundance of proteins or there are other reasons? This should be explained to readers, even because some protocols (MACSPlex exosome kit) rely on EV count.
Response:
As reported by reviewer 2, the NTA data can contain bias when applied to complex matrices such as blood and urines. Despite that, NTA is currently reported as a possible tool to quantify EVs together with the confirmation of other imaging-related techniques. Because of the relevant limitation highlighted by both Reviewers 1 and 3 in this calculation and the absence of TEM or ONI quantitative data on EV numbers, we preferred to eliminate this parameter from our analysis and revised manuscript. We rely on the use of EV numbers or proteins for MACplex analysis by applying a tetraspanin normalization factor that can bypass the problems associated with EV contamination. An extensive discussion on the limitations of NTA on urine analysis can be found in the cited position paper (ref. n.9) of the Urinary Task Force.
- Line 263 The sentence “In preliminary experiments, no significant modulation of miR21, miR24, miR221, miR296, and miR320c 264 was observed (Suppl. Table 1).” Apparently describe a PCR experiment, but actually in the supplementary material there is just a table with oligo used. The sentence and the Supplementary material should be coherent.
Response: We are deeply sorry for the confusion, the PCR data are reported on Suppl. Table 2.
- The sentence “At variance, the level of miR145 and miR126 in uEVs from diabetic patients appeared significantly modulated within the groups.” Is unclear. What do authors mean for “At variance”?
Response: The contrast of the data regarding miR145 and miR126 with the PCR data on the other miRNAs should be clear now, considering the correct reference to the data in Suppl. Table 2.
- In Figure 3B, the indication of PODO-Cells would help to catch its meaning
Response: We thank for the suggestion, we added the indication in the Figure.
Reviewer 3 Report
The paper is technically sound and the data support the conclusions. I have no further comments.
Author Response
We thank the reviewer for the positive comment.
Round 3
Reviewer 1 Report
The authors have satisfactorily addressed all of my concerns.
Reviewer 2 Report
The current version of the manuscript has amended its weak points